# Effects of *TIMP-2* Polymorphisms on Retinopathy of Prematurity Risk, Severity, Recurrence, and Treatment Response

**DOI:** 10.3390/ijms232214199

**Published:** 2022-11-17

**Authors:** Pei-Liang Wu, Xiao Chun Ling, Eugene Yu-Chuan Kang, Kuan-Jen Chen, Nan-Kai Wang, Laura Liu, Yen-Po Chen, Yih-Shiou Hwang, Chi-Chun Lai, Shun-Fa Yang, Wei-Chi Wu

**Affiliations:** 1Department of Medicine, National Taiwan University, Taipei 106, Taiwan; 2Department of Ophthalmology, Chang Gung Memorial Hospital, Linkou, Taoyuan 333, Taiwan; 3College of Medicine, Chang Gung University, Taoyuan 333, Taiwan; 4School of Traditional Chinese Medicine, Chang Gung University, Taoyuan 333, Taiwan; 5Department of Ophthalmology, Chang Gung Memorial Hospital, Tucheng, New Taipei City 236, Taiwan; 6Department of Ophthalmology, Chang Gung Memorial Hospital, Keelung 204, Taiwan; 7Institute of Medicine, Chung Shan Medical University, Taichung 402, Taiwan; 8Department of Medical Research, Chung Shan Medical University Hospital, Taichung 402, Taiwan

**Keywords:** *TIMP-2*, retinopathy of prematurity, single nucleotide polymorphisms (SNPs)

## Abstract

Tissue inhibitors of metalloproteinases (TIMPs) play a crucial role in endogenous angiogenesis besides the regulation of matrix metalloproteinase (MMP) activity. Associations between *TIMP-2* gene polymorphisms and the risk of retinopathy of prematurity (ROP) were examined. Premature infants born between 2009 and 2018 were included. Five single-nucleotide polymorphisms (SNPs) of *TIMP-2* were analyzed with real-time polymerase chain reaction (PCR). Multivariate logistic regression was applied to model associations between *TIMP-2* polymorphisms and ROP susceptibility and severity. The GA+AA genotype in individuals with the *TIMP-2* polymorphism of rs12600817 was associated with a higher risk of ROP (odds ratio [OR]: 1.518, 95% confidence interval [CI]: 1.028–2.242) compared with their wild-type genotypes. The AA genotype (OR: 1.962, 95% CI: 1.023–3.762) and the AA+GA genotype (OR: 1.686, 95% CI: 1.030–2.762) in individuals with the rs12600817 polymorphism had higher risks of severe, treatment-requiring ROP relative to their wild-type counterparts. In patients with treatment-requiring ROP, the AG+GG genotypes in the *TIMP-2* polymorphism of rs2889529 were correlated with the treatment response (*p* = 0.035). The *TIMP-2* polymorphism of rs12600817 help in predicting ROP risks in preterm infants, while the polymorphism of rs2889529 can serve as a genetic marker in evaluating the ROP treatment response.

## 1. Introduction

Retinopathy of prematurity (ROP) is a neovascular retinopathy that occurs among premature infants. Severe complications include retinal detachment, causing severe vision loss or blindness [1]. ROP has two phases: The first phase involves a delay of retinal vascular growth after birth, and the second phase comprises hypoxia-induced pathological vessel growth [2]. The pathophysiology of ROP is evolving toward genetic factors as the advancement of prenatal care improves the survival of premature babies.

Genetic polymorphism includes single-nucleotide polymorphism (SNP) [3], which refers to the presence of two distinct nucleotide alleles in genome positions that appear in a significant portion of the population [4]. SNPs have been used to predict diseases and individualize medical treatment. For example, they can be used to detect patients’ responses to drugs and chemicals and track genetic variance among family members [5]. 

The human *TIMP-2* gene (OMIM 188825) is located on chromosome 17q25 [6]. Tissue inhibitor of metalloproteinase 2 (TIMP-2) maintains tissue homeostasis [7] by suppressing the proliferation of quiescent tissues through angiogenic factors, according to the NCBI’s Reference Sequence (RefSeq) [6,8]. Most studies on *TIMP-2* polymorphisms have been performed in relation to cancers [9,10]. Given that neovascularization occurs in both cancers and ROP, studies have discussed the correlation between *TIMP-2* and ROP [11] or retinal neovascularization [12]. Still, most of them have focused on the relationship of TIMP-2 with matrix metalloproteinases (MMPs) rather than its direct effect on ROP.

Here, we examined five *TIMP-2* SNPs to evaluate whether *TIMP-2* SNPs affect ROP risk, severity, recurrence, and response to treatment.

## 2. Results

The demographics and clinical characteristics of the studied premature infants are presented in Table 1. Of the 450 premature infants examined, 224 had ROP (the ROP group), and 226 did not (the non-ROP group, which served as controls). GA and BW significantly differed between the non-ROP and ROP groups (*p* < 0.001). No significant difference was observed in sex (*p* = 0.775). In subsequent models that examined the risk profiles associated with genetic polymorphisms, GA and BW were controlled as confounders in each comparison of adjusted ORs and 95% CIs.

Table 2 presents the relationships between *TIMP-2* polymorphisms and the distribution of patients with or without ROP. In the study population, the *TIMP-2* alleles with the highest frequencies included rs2889529, rs8068674, rs16971783, rs7220980, and rs12600817, and each was homozygous for A/A, C/C, T/T, A/A, G/G. For the rs2889529, rs8068674, rs16971783, and rs7220980 polymorphisms, no significant difference was observed in the adjusted OR when comparing the ROP groups with their wild-type counterparts (Table 2). This finding indicated that the aforementioned polymorphisms were not associated with increased risks of ROP. For rs12600817, significant differences in percentage were found in AA+GA, but not GA or AA individually, compared with the wild-type genotypes (OR: 1.518, 95% CI: 1.028–2.242; *p* = 0.035). This result indicated higher risks of ROP for the AA+GA genotype of the rs12600817 polymorphism than for the wild-type genotypes.

Next, the ROP group (*n* = 224) was subdivided into mild ROP (*n* = 114) and severe ROP (*n* = 110). We then discussed the relationship between their *TIMP-2* polymorphisms and the severity of ROP (Table 3). No significant differences in the percentage of ROP patients were observed in the rs2889529, rs8068674, rs16971783, and rs7220980 polymorphisms in either subgroup compared with their wild-type counterparts. No significant difference in the percentage of ROP patients was found in the rs12600817 polymorphism between the mild ROP group and the wild-type counterpart. However, the AA (OR: 1.962, 95% CI: 1.023–3.762; *p* = 0.043) and AA+GA (OR: 1.686, 95% CI: 1.030–2.762; *p* = 0.037) genotypes showed significant differences in percentage between the severe ROP group and the wild-type counterpart (Table 3). This result indicated that both the AA and AA+GA genotypes were associated with higher risks of severe ROP than their corresponding wild-type genotypes.

Next, we investigated the relationship between ROP and the frequency of *TIMP-2* alleles. Among the 450 patients recruited, a total of 900 alleles were recorded. Of the 900 alleles, 452 belonged to the non-ROP group, while 448 belonged to the ROP group (Table 4). The rs2889529, rs8068674, rs16971783, and rs7220980 polymorphisms did not exhibit a significant percentage difference compared with the wild-type counterpart. For rs12600817, the G and A alleles showed a significant percentage difference (OR: 1.336, 95% CI: 1.025–1.743; *p* = 0.032), which indicated that patients with the A allele of the rs12600817 polymorphism had a higher ROP risk.

Then, the ROP conditions were further separated into mild and severe ROP. According to Table 5, 228 alleles belonged to the mild ROP group, while 220 were in the severe ROP group. The rs2889529, rs8068674, rs16971783, and rs7220980 polymorphisms did not exhibit significant differences in percentage in either subgroup compared with the wild-type counterparts. No significant difference in percentage was observed in the rs12600817 polymorphism between the mild ROP group and the non-ROP group, but the severe ROP group exhibited a significant percentage difference compared with the wild-type counterpart (OR: 1.446, 95% CI: 1.044–2.002; *p* = 0.026). The results indicated that the A allele of the rs12600817 polymorphism was associated with a higher risk of severe ROP in premature infants.

Finally, rs2889529 and its association with the clinicopathologic characteristics of ROP were examined (Table 6). Patients with severe ROP carrying the AG+GG genotype had a significantly lower chance and risk of progressing from stage 3 to 5 ROP (*p* = 0.010) compared with the wild-type counterparts. In addition, no significant difference in percentage was observed in ROP recurrence between patients with the AA genotype and those with the AG and GG genotypes (*p* = 0.198) or the subset of patients with severe ROP (*p* = 0.157). Compared with the wild-type AA genotype, the AG+GG genotype was associated with a significantly decreased treatment response for all the patients (OR: 0.404, 95% CI: 0.173–0.943; *p* = 0.032). Furthermore, in the severe ROP group, patients with the AG+GG genotype showed significantly worse treatment responses than those with the wild-type AA genotype (OR: 0.383, 95% CI: 0.154–0.952; *p* = 0.035).

## 3. Discussion

This study evaluated the relationship between different *TIMP-2* polymorphisms and their ROP risks, treatment response, and ROP recurrence. The main findings of the study are that the rs12600817 polymorphism of *TIMP-2* could be an indicator of the risks of ROP in premature infants. Patients with at least one A allele in the genotype of the rs12600817 polymorphism exhibited a higher ROP risk than their homozygous GG counterpart (OR: 1.686, 95% CI: 1.030–2.762; *p* = 0.037). In addition, the rs2889529 polymorphism can be used to evaluate the ROP treatment response. The presence of one or two G alleles in the genotype was associated with a lower risk of stage 3–5 ROP than their homozygous AA counterparts (*p* = 0.010). Patients with the G allele had a worse treatment response for ROP than those with the AA genotype, indicating that the G allele can serve as a genetic marker of the ROP treatment response.

The TIMP-2 protein acts as a metastasis suppressor [13,14,15] and can inhibit the reaction of MMPs, such as MMP-2 [14,16] and MMP-9 [16]; thus, it plays a role in the maintenance of the extracellular matrix of the retina [17,18]. It can also inhibit the mitogenic response of human microvascular endothelial growth factor, which can lead to endothelial cell proliferation [7], the suppression of which can inhibit angiogenesis. ROP is caused by the abnormal proliferation of blood vessels, eventually progressing to retinal detachment. Our data indicated that the *TIMP-2* polymorphism of rs12600817, which contains the A allele, might cause the loss of function of the TIMP-2 protein, inhibiting angiogenesis, causing unnormal vessel proliferation, and eventually leading to ROP development.

Our findings demonstrate *TIMP-2* variation as a novel factor of ROP compared with traditional risk factors, such as GA and BW. Although gene therapy for SNPs is not yet available, a better understanding of genes and their influence on downstream factors can help devise better treatment strategies for ROP. SNP testing can help in risk prediction in individuals in the future. For example, identifying the rs12600817 polymorphism can lead to testing for the A allele, thereby enabling the risk prediction of ROP in premature infants. Similarly, identifying the G allele of the rs2889n529 polymorphism can indicate a lower potential response to ROP treatment, which may serve as a treatment marker.

Consistent with previous studies, decreased *TIMP-2* expression may lead to myopia in mammals [19] and did have an effect on breast cancer [20]. Unlike the traditional view that TIMP-2 is considered an MMP inhibitor, ours is the first study to investigate the *TIMP-2* genome and identify that the rs12600817 polymorphism can affect the development of severe ROP in premature infants. In addition, the G allele in the rs2889529 polymorphism seemed to have two effects on patients with severe ROP. First, it appeared to decrease the risks of stage 3–5 ROP in patients. Second, the data suggested that the G allele tended to reduce patients’ treatment responses.

Hypoxia-Induced retinal neovascularization is often observed in the second phase of ROP [2]. As former studies have shown, *VEGF* has been linked to neovascularization [21], while hypoxia served as a gene inducer for VEGF [22]. Therefore, activation of VEGF had long been linked to the second phase of ROP. On the other hand, the *TIMP-2* gene has been shown to phosphorylate the VEGF receptor, which causes the VEGF receptor to be inhibited [23]. The rs12600817 polymorphism could potentially alter the function of TIMP-2, which could no longer cause phosphorylation of the VEGF receptor. As the VEGF receptor is no longer inhibited, a large amount of VEGF could be expressed, leading to neovascularization and causing severe treatment needing ROP.

With these newly found results, the diagnosis and treatment response for ROP could be improved clinically. For preterm infants, polymorphism rs12600817 could serve as a biomarker for severe ROP. For infants with genotypes AA and GA, closer follow-up of the infant’s eye condition should be exercised to ensure a timely treatment is performed on time once patients’ retinopathy worsens. Polymorphism rs2889529 should also be checked because it can serve as a genetic marker for treatment response. A worse response to the treatment was expected for infants with AG and GG genotypes. Thus, more aggressive or combined interventions of various treatments could be considered in such patients.

This study has a few limitations. First, the study population enrolled for this study was large compared with that in other ROP SNP studies [24,25,26,27,28], but the number is small compared with studies of SNPs in other visual diseases [29,30,31,32,33]. Second, because the study was conducted in Taiwan, all of the patients were of East Asian/Han Chinese ethnicity; thus, the study results may not be generalizable to other ethnicities; studies have shown that infants from Asian countries have higher risks of ROP than white infants [34,35]. Third, we only studied *TIMP-2*, but other genes could also influence the occurrence of ROP and its severity. If this is true, it might indicate that ROP is a multigene disease, in which *TIMP-2* only contributed to a portion of the risk to ROP. Fourth, the analysis of different study groups did not include sex, which can affect ROP. Other maternal factors such as maternal age [36], hypertensive disorders of pregnancy, and maternal diabetes mellitus may exert effects, but they were not investigated here [37]. Future studies should investigate how these factors and genotypes influence the expression of ROP and whether the different SNPs of *TIMP-2* affect the condition of patients with ROP.

## 4. Materials and Methods

### 4.1. Study Population

This study was conducted at the Linkou and Taipei branches of Chang Gung Memorial Hospital, Taiwan. This study was approved by the Institutional Review Board of Chang Gung Medical Foundation (No. 202001715A3) and adhered to the tenets of the Declaration of Helsinki.

Infants born between 2009 and 2018 were enrolled after written informed consent was obtained from their parents. Both full-term infants and preterm infants (defined as those born before 37 weeks of gestational age [GA] or with birth weight [BW] < 1500 g) were included [38]. All of the infants were born in Taiwan and had an ethnicity of Chinese. We excluded patients without complete medical records or with <6-month follow a period. In total, 55 patients were excluded due to below reasons: 14 patients had incomplete medical records because they were born in other medical centers; 8 patients withdrew from the study due to long-hour of waiting for the detailed eye exam; 33 patients withdrew because they were afraid of the blood drawing process. All included infants underwent screening for ROP that was conducted by two ophthalmologists. Depending on whether the screening criteria for ROP were met, the infants were divided into no ROP and ROP groups. All the patients were first examined by doctors with binocular examination for the grading of ROP according to the criteria of ICROP [39]. RetCam was used for the documentation of the progression of the disease and is generally reserved for more severe cases. Based on their condition, ROP was subdivided into mild (no treatment required, including type 2, or milder than type 2) and severe (type 1) ROP [40]. Type 1 ROP, for which treatment (either anti-vascular endothelium growth factor or laser photocoagulation) is indicated, was defined as zone I, any stage ROP with plus disease (a degree of dilation and tortuosity of the posterior retinal blood vessels meeting or exceeding that of a standard photograph); zone I, stage 3 ROP without plus disease; or zone II, stage 2 or 3 ROP with plus disease. There were 112 patients grouped as treatment-requiring ROP. Among the treated eyes, 80 patients received IVI of anti-VEGF (56 with bevacizumab, 8 with ranibizumab, 16 aflibercept), 12 patients received laser photocoagulation, and 20 received both IVI of anti-VEGF and laser photocoagulation. Type 2 ROP was defined as ROP for which treatment was not indicated but required close clinical monitoring, which included zone I, stage 1 or 2 ROP without plus disease or zone II, and stage 3 ROP without plus disease [40].

Recurrence after treatment with IVR was defined as having initial ROP regression, followed by the reappearance of plus disease, preretinal and vitreous hemorrhage, worsening of retinal neovascularization, or progression to retinal detachment. Nonresponders were defined as those cases with persistence or worsening of plus disease, persistence or worsening of neovascular proliferation, or progression to retinal detachment [41,42,43].

### 4.2. Selection of TIMP-2 Polymorphism

More than 29,321 SNPs have been documented in the dbSNPs database regarding the intron or downstream of the *TIMP-2* gene region. Overall, the selection of SNPs for association analysis of candidate genes adhered to the following principles: (1) Haplotype-tagging (htSNPs) SNPs are derived from the analyses of HapMap data and essentially provide a minimal set of markers that would guarantee that at least one marker is in strong linkage disequilibrium with any unmeasured marker according to our estimates of pairwise linkage disequilibrium measures. (2) The most known polymorphisms of the coding region (cSNPs) were included in genotyping. Ultimately, we selected five *TIMP-2* SNPs—rs2889529, rs8068674, rs16971783, rs7220980, and rs12600817—because they adhered to the aforementioned principles and were reported in studies on *TIMP-2* pathophysiology in ocular diseases [44,45,46].

### 4.3. Genomic Data Extraction

The patients’ DNA was extracted using a DNA collection kit (Oragene-DNA; DNA-Genotek, Ottawa, ON, Canada) with 3 mL blood samples as per the manufacturers’ instructions. The extracted DNA samples were dissolved in TE buffer (10 mM Tris at pH 7.8 and 1 mM EDTA), and the optical density of the absorbance was measured at 260 nm, which was measured by Thermo Scientific^TM^ NanoDrop 2000 UV-Vis spectrophotometers (Thermo Fisher Scientific, Waltham, MA, USA). The products were then stored at −20 °C to create templates for polymerase chain reaction (PCR)

### 4.4. Real-Time PCR

The rs2889529, rs8068674, rs16971783, rs7220980, and rs12600817 variants of *TIMP-2* were assessed using the ABI StepOne Real-Time PCR System (Applied Biosystems, Foster City, CA, USA) and analyzed using SDS v3.0 software (Applied Biosystems) with a TaqMan assay. Each reaction required a mixture of 2.5 mL of TaqMan Genotyping Master Mix, 0.125 mL of the TaqMan probe mix, and 10 ng of genomic DNA, with a final volume of 5 mL. The initial denaturation was performed at 95 °C for 10 min, followed by 40 cycles of 95 °C for 15 s and 60 °C for 1 min for annealing and extension.

### 4.5. Statistical Analyses

Hardy–Weinberg equilibrium was assessed using a goodness-of-fit χ^2^ test for biallelic markers. The ROP characteristics between the ROP and non-ROP groups were compared using the Mann–Whitney U test and Fisher’s exact test. The multivariate logistic regression model was applied to study the associations between the *TIMP-2* variants and ROP susceptibility and their effect on ROP severity. We then used multiple logistic regression models to estimate the risks and clinicopathological characteristics by calculating the adjusted odds ratios (ORs) and 95% confidence intervals (CIs) associated with the genotype frequencies. The differences were considered significant at *p* < 0.05. All statistical analyses were performed using SAS software version 9.1 (SAS Institute, Cary, NC, USA).

## 5. Conclusions

There were two main findings from this study. First, the *TIMP-2* polymorphism of rs12600817 can help in the prediction of ROP risk in preterm infants. Second, the *TIMP-2* polymorphism of rs2889529 can serve as a genetic marker for evaluating the ROP treatment response. This information can guide the individualized management and treatment of patients with ROP.

## Figures and Tables

**Table 1 ijms-23-14199-t001:** Demographic and clinical characteristics of infants with and without retinopathy of prematurity.

Subject Characteristics	Non-ROP (*n* = 226)	ROP (*n* = 224)	*p* Value
**Gestational age (weeks)**			
Mean ± SD (weeks)	31.27 ± 2.81	26.95 ± 2.15	<0.001
**Birth weight (gram)**			
Mean ± SD (weeks)	1521.70 ± 517.60	883.86 ± 261.29	<0.001
**Gender, n (%)**			
Male	115 (50.9%)	117 (52.2%)	0.775
Female	111 (49.1%)	107 (47.8%)	
**Stage**			
1 + 2		113 (50.4%)	
3 + 4 + 5		111 (49.6%)	
**Zone**			
1 + 2		204 (91.1%)	
3		20 (8.9%)	
**Plus**			
0		133 (59.4%)	
1		91 (40.6%)	
**Recurrent**			
No		216 (96.4%)	
Yes		8 (3.6%)	
**Response**			
No		27 (12.1%)	
Yes		197 (87.9%)	

**Table 2 ijms-23-14199-t002:** Distribution frequency of *TIMP-2* genotypes in infants with and without retinopathy of prematurity.

Genotype SNP	Non-ROP (*n* = 226)	ROP (*n* = 224)	OR (95% CI)	*p* Value
**rs2889529**				
AA	109 (48.2%)	118 (52.7%)	1.00	
AG	94 (41.6%)	82 (36.6%)	0.806 (0.543–1.195)	0.283
GG	23 (10.2%)	24 (10.7%)	0.964 (0.514–1.807)	0.909
AG+GG	117 (51.8%)	106 (47.3%)	0.837 (0.578–1.212)	0.345
**rs8068674**				
CC	117 (51.8%)	113 (50.4%)	1.00	
CT	92 (40.7%)	94 (42.0%)	1.058 (0.719–1.557)	0.775
TT	17 (7.5%)	17 (7.6%)	1.035 (0.504–2.128)	0.925
CT+TT	109 (48.2%)	111 (49.6%)	1.054 (0.729–1.526)	0.779
**rs16971783**				
TT	196 (86.7%)	190 (84.8%)	1.00	
TA	30 (13.3%)	31 (13.8%)	1.066 (0.621–1.830)	0.817
AA	0 (0.0%)	3 (1.4%)	---	
TA+AA	30 (13.3%)	34 (15.2%)	1.169 (0.688–1.986)	0.563
**rs7220980**				
AA	161 (71.2%)	146 (65.2%)	1.00	
AG	61 (27.0%)	71 (31.7%)	1.284 (0.853–1.932)	0.232
GG	4 (1.8%)	7 (3.1%)	1.930 (0.554–6.727)	0.302
AG+GG	65 (28.8%)	78 (34.8%)	1.323 (0.889–1.970)	0.167
**rs12600817**				
GG	90 (39.8%)	68 (30.4%)	1.00	
GA	99 (43.8%)	109 (48.6%)	1.457 (0.961–2.209)	0.076
AA	37 (16.4%)	47 (21.0%)	1.681 (0.986–2.867)	0.056
GA+AA	136 (60.2%)	156 (69.6%)	1.518 (1.028–2.242)	0.035 *

Abbreviations: SNP, single-nucleotide polymorphism; OR, odds ratio; CI: confidence interval. * Statistically-significant at *p* < 0.05.

**Table 3 ijms-23-14199-t003:** Distribution frequency of *TIMP-2* genotypes in infants without or with mild or severe retinopathy of prematurity.

SNP Genotypes	Non-ROP (*n* = 226)	Mild ROP		Severe ROP	
		(*n*= 114)	OR (95% CI)	(*n* = 110)	OR (95% CI)
**rs2889529**					
AA	109 (48.2%)	62 (54.4%)	1.00	56 (50.9%)	1.00
AG	94 (41.6%)	38 (33.3%)	0.711 (0.436–1.159)	44 (40.0%)	0.911 (0.563–1.475)
GG	23 (10.2%)	14 (12.3%)	1.070 (0.514–2.229)	10 (9.1%)	0.846 (0.377–1.901)
AG+GG	117 (51.8%)	52 (45.6%)	0.781 (0.497–1.227)	54 (49.1%)	0.898 (0.569–1.417)
**rs8068674**					
CC	117 (51.8%)	55 (48.2%)	1.00	58 (52.7%)	1.00
CT	92 (40.7%)	47 (41.2%)	1.087 (0.675–1.749)	47 (42.7%)	1.031 (0.643–1.652)
TT	17 (7.5%)	12 (10.6%)	1.502 (0.671–3.360)	5 (4.6%)	0.593 (0.209–1.688)
CT+TT	109 (48.2%)	59 (51.8%)	1.151 (0.734–1.807)	52 (47.3%)	0.962 (0.610–1.519)
**rs16971783**					
TT	196 (86.7%)	97 (85.1%)	1.00	93 (84.5%)	1.00
TA	30 (13.3%)	15 (13.2%)	1.010 (0.519–1.966)	16 (14.6%)	1.124 (0.584–2.164)
AA	0 (0.0%)	2 (1.7%)	---	1 (0.9%)	-
TA+AA	30 (13.3%)	17 (14.9%)	1.145 (0.602–2.178)	17 (15.5%)	1.194 (0.627–2.274)
**rs7220980**					
AA	161 (71.2%)	74 (64.9%)	1.00	72 (65.5%)	1.00
AG	61 (27.0%)	36 (31.6%)	1.284 (0.782–2.107)	35 (31.8%)	1.283 (0.778–2.115)
GG	4 (1.8%)	4 (3.5%)	2.176 (0.530–8.938)	3 (2.7%)	1.677 (0.366–7.687)
AG+GG	65 (28.8%)	40 (35.1%)	1.339 (0.828–2.165)	38 (34.5%)	1.307 (0.803–2.128)
**rs12600817**					
GG	90 (39.8%)	37 (32.5%)	1.00	31 (28.2%)	1.00
GA	99 (43.8%)	55 (48.2%)	1.351 (0.815–2.239)	54 (49.1%)	1.584 (0.936–2.679)
AA	37 (16.4%)	22 (19.3%)	1.446 (0.754–2.776)	25 (22.7%)	1.962 (1.023–3.762) ^a^
GA+AA	136 (60.2%)	77 (67.5%)	1.377 (0.857–2.212)	79 (71.8%)	1.686 (1.030–2.762) ^b^

^a^*p* = 0.043; ^b^
*p* = 0.037.

**Table 4 ijms-23-14199-t004:** Distribution frequency of *TIMP-2* alleles of infants with and without retinopathy of prematurity.

Genotype SNP	Non-ROP (*n* = 452)	ROP (*n* = 448)	OR (95% CI)	*p* Value
**rs2889529**				
A allele	312 (69.0%)	318 (71.0%)	1.00	
G allele	140 (31.0%)	130 (29.0%)	0.911 (0.685–1.212)	0.522
**rs8068674**				
C allele	326 (72.1%)	320 (71.4%)	1.00	
T allele	126 (27.9%)	128 (28.6%)	1.035 (0.774–1.384)	0.817
**rs16971783**				
T allele	422 (93.4%)	411 (91.7%)	1.00	
A allele	30 (6.6%)	37 (8.3%)	1.266 (0.768–2.089)	0.354
**rs7220980**				
A allele	383 (84.7%)	363 (81.0%)	1.00	
G allele	69 (15.3%)	85 (19.0%)	1.300 (0.917–1.842)	0.140
**rs12600817**				
G allele	279 (61.7%)	245 (54.7%)	1.00	
A allele	173 (38.3%)	203 (45.3%)	1.336 (1.025–1.743)	0.032 *

Abbreviations: SNP, single-nucleotide polymorphism; OR, odds ratio; CI: confidence interval. * Statistically-significant at *p* < 0.05.

**Table 5 ijms-23-14199-t005:** Distribution frequency of *TIMP-2* alleles of infants without and with mild or severe retinopathy of prematurity.

SNP Genotypes	Non-ROP (*n* = 452)	Mild ROP		Severe ROP	
		(*n* = 228)	OR (95% CI)	(*n* = 220)	OR (95% CI)
**rs2889529**					
A allele	312 (69.0%)	162 (71.1%)	1.00	156 (70.9%)	1.00
G allele	140 (31.0%)	66 (28.9%)	0.908 (0.641–1.287)	64 (29.1%)	0.914 (0.643–1.301)
**rs8068674**					
C allele	326 (72.1%)	157 (68.9%)	1.00	163 (74.1%)	1.00
T allele	126 (27.9%)	71 (31.1%)	1.170 (0.826–1.656)	57 (25.9%)	0.905 (0.628–1.303)
**rs16971783**					
T allele	422 (93.4%)	209 (91.7%)	1.00	202 (91.8%)	1.00
A allele	30 (6.6%)	19 (8.3%)	1.279 (0.703–2.326)	18 (8.2%)	1.253 (0.682–2.302)
**rs7220980**					
A allele	383 (84.7%)	184 (80.7%)	1.00	179 (81.4%)	1.00
G allele	69 (15.3%)	44 (19.3%)	1.327 (0.875–2.014)	41 (18.6%)	1.271 (0.831–1.945)
**rs12600817**					
G allele	279 (61.7%)	129 (56.6%)	1.00	116 (52.7%)	1.00
A allele	173 (38.3%)	99 (43.4%)	1.238 (0.896–1.710)	104 (47.3%)	1.446 (1.044–2.002) ^a^

^a^*p* = 0.026.

**Table 6 ijms-23-14199-t006:** Clinicopathologic characteristics of infants with retinopathy of prematurity, stratified by polymorphic genotypes of *TIMP-2* rs2889529.

Variable	ALL (*n* = 224)			Severe ROP (*n* = 110)		
	AA (*n* = 118)	AG+GG (*n* = 106)	*p* Value	AA (*n* = 56)	AG+GG (*n* = 54)	*p* Value
**Stage**						
1 + 2	59 (50.0%)	54 (50.9%)	*p* = 0.888	0 (0.0%)	6 (11.1%)	*p* = 0.010
3 + 4 + 5	59 (50.0%)	52 (49.1%)		56 (100%)	48 (88.9%)	
**Zone**						
1 + 2	104 (88.1%)	100 (94.3%)	*p* = 0.104	56 (100%)	54 (100%)	---
3	14 (11.9%)	6 (5.7%)		0 (0.0%)	0 (0.0%)	
**Plus**						
0	73 (61.9%)	60 (56.6%)	*p* = 0.423	11 (19.6%)	8 (14.8%)	*p* = 0.503
1	45 (38.1%)	46 (43.4%)		45 (80.4%)	46 (85.2%)	
**Recurrent**						
No	112 (94.9%)	104 (98.1%)	*p* = 0.198	50 (89.3%)	52 (96.3%)	*p* = 0.157
Yes	6 (5.1%)	2 (1.9%)		6 (10.7%)	2 (3.7%)	
**Response**						
No	9 (7.6%)	18 (17.0%)	*p* = 0.032 ^a^	9 (16.1%)	18 (33.3%)	*p* = 0.035 ^b^
Yes	109 (92.4%)	88 (83.0%)		47 (83.9%)	36 (66.7%)	

^a^ OR (95% CI): 0.404 (0.173–0.943); ^b^ OR (95% CI): 0.383 (0.154–0.952).

## Data Availability

The data presented in this study are available on request from the corresponding author. The data are not publicly available due to privacy.

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
