# Peer review of "Effects of *TIMP-2* Polymorphisms on Retinopathy of Prematurity Risk, Severity, Recurrence, and Treatment Response"

_ijms, 2022, doi:10.3390/ijms232214199_

Round 1

Reviewer 1 Report

There were two findings for the study. First, the TIMP-2 polymorphism of rs12600817 can help for prediction of ROP risks in preterm infants. Second, the TIMP-2 polymorphism of rs2889529 can serve as a genetic marker for evaluating the ROP treatment response. I agree this is important and interesting but from my point a view. I would change to Could Effects of TIMP-2 Polymorphisms on Retinopathy of Prematurity-Risk, Severity, Recurrence, and Treatment Response. Sorry to say that I man not convinced

• The author describe different polymorphisms in metalloproteinases (TIMPs) and happen to find higher risk for rop for one poymorphism, that need treatment. P 0.032.

• Were all children from the same geographic area, as genetic risk factor are depending on that.

• As some seems to be treated, what type of treatment was performed.

• Grading of ROP was that by retcam or several doctors with binocular examination

• According to the manuscript were same were excluded because of different reason (without complete medical records). ??? number of patients and reason

Reviewer 2 Report

This manuscript does not have Methodology

Please include and resubmit

Reviewer 3 Report

In this manuscript, the author studies the relationship between different TIMP-2 polymorphisms and the risks of ROP. I don’t have too much comments on the manuscript. However, I hope the authors could give some comments on following 2 questions.

1.    According to the introduction, ROP typically has 2 phases. The early phase of premature of retina blood vessels, and the following abnormal retinal NV generation. Since in this study, the author concluded that the rs12600817 polymorphism can be a potential indicator of ROP, please make some comments about how the rs12600817 influences different phases of the disease development.

2.    Also, how we can take use of the discovered biomarker for ROP diagnose and/or treatment.

Round 2

Reviewer 1 Report

The manuscript has improved

Reviewer 2 Report

This is a well written manuscript highlighting TIMP - 2 Polymorphism which be useful addition to literature.